# Adaptive High-Resolution Imaging Method Based on Compressive Sensing

**DOI:** 10.3390/s22228848

**Published:** 2022-11-16

**Authors:** Zijiao Wang, Yufeng Gao, Xiusheng Duan, Jingya Cao

**Affiliations:** 1School of Mechanical Engineering, Shijiazhuang Tiedao University, Shijiazhuang 050000, China; 2School of Engineering, Hong Kong University, Hong Kong, China; 3School of Artificial Intelligence and Big Data, Hebei Polytechnic Institute, Shijiazhuang 050000, China

**Keywords:** compressive sensing, adaptive imaging, fiber array, high pixels

## Abstract

Compressive sensing (CS) is a signal sampling theory that originated about 16 years ago. It replaces expensive and complex receiving devices with well-designed signal recovery algorithms, thus simplifying the imaging system. Based on the application of CS theory, a single-pixel camera with an array-detection imaging system is established for high-pixel detection. Each detector of the detector array is coupled with a bundle of fibers formed by fusion of four bundles of fibers of different lengths, so that the target area corresponding to one detector is split into four groups of target information arriving at different times. By comparing the total amount of information received by the detector with the threshold set in advance, it can be determined whether the four groups of information are calculated separately. The simulation results show that this new system can not only reduce the number of measurements required to reconstruct high quality images but can also handle situations wherever the target may appear in the field of view without necessitating an increase in the number of detectors.

## 1. Introduction

As a type of signal sampling theory, CS theory overcomes the shortcomings of traditional Nyquist theorem sampling, namely a large amount of data and a large waste of storage space and time. Researchers proposed that if the original signal is sparse or compressible, a precise reconstruction can be recovered from a small number of random measurements by solving linear equations [1,2,3,4]. Since the non-adaptive linear measurement results of a small number of compressible signals or images contain enough information for reconstruction processing, CS theory can be used to directly obtain the compressed signal representation without first sampling the signal [4,5,6,7]. CS theory transfers the burden of sampling to data processing, it shifts the focus of the imaging system from the traditional design of expensive receiver hardware to the novel design of signal recovery algorithms [5,7]. At present, CS technology has been widely used in three areas—wireless communication, array signal processing, and imaging systems—and plays a major role in fault diagnosis and signal recognition [8,9]. A single-pixel camera is an important application of compressed sensing theory. It uses a digital micromirror device (DMD) to perform optical calculations on the linear projection of images on the pseudo-random binary pattern [10]. Single-pixel camera technology has been widely used in spectral imaging, radar imaging, and medical imaging [11,12]. Similarly, it also supports a variety of applications for three-dimensional (3-D) imaging, such as multispectral imaging [13], depth estimation of imaging scenes [14], and video rate improvement [15].

In previous research [16], the sparsity problem was identified, which has a great impact on array detection imaging systems based on CS theory; consequently, a method using irregularly arranged fibers was proposed to solve this problem. However, this method should be applied under a certain condition, i.e., the target should always be in the middle of the field of view. The reconstruction of an image using fibers with fewer data is a hot research topic; one method is proposed by Yu et al. [17]. However, our method is based on CS theory rather than machine learning algorithm, and uses fewer datasets. Another related method is to conjunct the photon-counting imaging system with a fiber optic taper to extend the field of view of the images [18]. Therefore, in this article, taking advantage of the flexibility of the fiber array arrangement, the threshold judgment method is used to determine the position of the target, which facilitates the rapid completion of the pixels imaging process without increasing the number of detectors.

## 2. Theory

### 2.1. CS and Single-Pixel Camera

The CS theory enables researchers to stably reconstruct the image with fewer measurement results than the number of reconstructed pixels [7]. It comprises three main parts: sparse representation of signal, measurement matrix and reconstruction algorithm.

Figure 1 shows the principle of CS. If the *N*-dimensional signal *x* is *K*-sparse on the Ψ transform domain, it will be processed in the Φ domain. The measurement matrix y can be obtained after M linear measurements by:(1)y=ΦΨα=Θα
where Θ is called the effective observation matrix. An accurate or approximate solution can be obtained by the optimal norm using a small number of measured values y, so that x=Ψα has a higher probability of being reconstructed accurately. The process of signal reconstruction faces a typical complexity non-deterministic polynomial (NP) problem. According to CS theory, if the condition of restricted isometry property (RIP) is met [3], it can be converted to the $L_{1}$ norm to reconstruct *x*, and can be written as:(2)$$\begin{array}{c} \alpha^{*}=\arg\underset{\alpha }{\min}{\left\|\alpha \right\|}_{1}\\ s.t.\;{\left\|y-\Theta \alpha \right\|}_{2}\leq \varepsilon \end{array}$$
where ε is the maximum noise energy limit during reconstruction.

The single-pixel camera is an optical computer that sequentially measures the inner products $y[m]=\langle x,\phi_{m}\rangle$ between an *N*-pixel sampled version x of the incident light-field from the scene under view and a set of two-dimensional (2-D) test functions $\left\{\phi_{m}\right\}$ [9]. The structure of the single-pixel CS camera is shown in Figure 2.

The light-field is focused by biconvex Lens 1 onto a DMD consisting of an array of N small mirrors. Each mirror corresponds to a particular pixel in x, and $\phi_{m}$ can independently rotate in the direction controlled by the codes. The reflected light is then collected by biconvex Lens 2 and focused onto a single photon detector that integrates the product $x[n]\phi_{m}[n]$ to compute the measurement $y[m]=\langle x,\phi_{m}\rangle$ as its output voltage and digitized by the A/D converter. Values of $\phi_{m}$ can be obtained by dithering the mirrors back and forth during the photodiode integration time [7].

### 2.2. Problem

Baraniuk et al. proposed that if the relationship between the number of random measurements *M* and the *K*-sparse vectors satisfies
(3)M≥O(Klog(N/K))
these compressible vectors can be exactly reconstructed and approximated stably with high probability [19]. That is to say, at least *M* measurements are required to reconstruct a *K*-sparse signal with high quality. Among them, sparsity can be simply understood as the number of non-zero elements in the signal information, and values close to zero are sometimes approximated to zero. Large coefficients generally contain signal-related information, while sparse signals are small coefficients (such as background) that contain little information.

For example, a single-pixel CS camera is used to image a target and reconstruct an image of 256 × 256 pixels (65,536 pixels), in which background information occupies more than 70%, and a high-quality reconstructed image that meets the recognition needs may require approximately 1700 random measurements. This reflects the advantage of CS theory; that is, compared with 256 × 256 detector array imaging, it not only reduces the detection cost but also shortens the detection time compared with the traditional single-pixel detection. However, in actual detection, more and more complex scenes or fine targets need to be captured, sampled, and processed faster in 3D imaging systems, with lower power consumption and larger detection pixels. Therefore, a compromise solution was proposed in previous studies: an array detector with a small number of pixels and a DMD array are used to image the target. Each detector can be regarded as a single-pixel camera, using compressed sensing theory to reconstruct the corresponding sub-image block to achieve the purpose of reducing the reconstruction time through parallel measurement and calculation. In the array detection imaging system, assuming that a 4 × 4 pixel array detector is used to detect the target to reconstruct a high-quality image of 256 × 256 pixels, the sub-imaging block corresponding to each detector is 64 × 64 pixels (4096 pixels). This method of image cutting results in several sub-imaging blocks that may be full of target information (close to 0-sparse), and the number of measurements calculated by Equation (3) is close to 4096 pixels, which removes the advantage of the CS technology in the array imaging system. Simultaneously, there may be several sub-imaging blocks that are almost full of background information, resulting in redundant measurement work. In previous studies, assuming that the target is always in the middle of the field of view and the background information is distributed around, it is proposed that the number of pixels of the image block corresponding to the target and the background information can be redistributed using an optical fiber arrangement, thereby shortening the high-quality reconstruction time for low-sparse sub-image blocks, and the redundant calculations of high-sparse sub-image blocks reconstruction are reduced [16]. However, the target may appear anywhere in the actual field of view, as shown in Figure 3. In this case, the previous optical fiber arrangement method cannot be used.

### 2.3. Proposed System

For this situation where the target position is not fixed, the structure of an adaptive imaging system is proposed, as shown in Figure 4 and Figure 5.

The pulsed beam is emitted by the laser and reflected from the target. The reflected signal reaches the DMD array board after passing through lens 1, and then is focused on the optical fiber array by lens 2. The DMD array board is composed of DMD boards arranged in 6 × 6, and each DMD board is composed of small mirrors arranged in 100 × 100 (that is, the final reconstructed target image is 600 × 600 pixels). At the input of the optical fiber array, 36 groups of optical fiber bundles are closely arranged in a 6 × 6 arrangement. Each group of optical fiber bundles consists of four optical fibers, and the lengths of these four optical fibers follow an arithmetic sequence. The tolerance ∆*L* can ensure that the time length of the signals returning from the same sub-region of the target do not overlap. The output of the four optical fibers is fused into one optical fiber and connected to a detector in the detector array, and the 36 detectors in the detector array are also arranged in a 6 × 6 arrangement. It is equivalent to each detector receiving the signal reflected by its corresponding one DMD board. The computer first judges the amount of signal received according to a preset threshold. If the total signal amount of one 100 × 100 pixels sub-image area exceeds the threshold within a certain period, the signal of the four time periods of this sub-image area is respectively measured and processed in parallel using CS theory. The reconstructed four 50 × 50 pixel images are finally spliced into a 100 × 100 pixel reconstructed image. On the contrary, when the total amount of signal is less than the threshold, the information of this sub-image area does not need to be processed separately, but can be directly used to reconstruct a 100 × 100 pixel image. Finally, all the reconstructed images are spliced together in the correct position to obtain a reconstructed image of 600 × 600 pixels.

It can be concluded from the previous analysis that when more information of the target is processed, longer measurement time is required for high-quality reconstruction. Therefore, the threshold is used to determine the location of the target, the sub-image blocks with target information greater than the threshold are further divided into four parts, and the measurement time is reduced through parallel calculation to achieve the purpose of shortening the overall detection and imaging time.

## 3. Comparative Experiment

In order to better demonstrate the advantages of the proposed system, a simulation system is built, and a series of simulation experiments are performed. Simulation parameters mainly include laser radar (LADAR) ranging equation, fiber loss, detector conversion efficiency, etc. The LADAR range equation describes the relevant parameters of the radar system, target characteristics, receiving parameters, and atmospheric transmission [20,21]. In the following simulation experiments, it is assumed: that the laser transmission in the atmosphere conforms to the geometrical optics principle; the atmosphere is uniform and isotropic; the target object belongs to the Lambert radiator; and the laser energy distribution on the surface of the target object is uniform [18]. The equation to compute signal power at the detector $P_{\det}$ that incorporates these efficiency terms is expressed by:(4)$$P_{\det}=\frac{\tau_{0}\tau_{a}D_{R}^{2}\rho_{t}(dA)P_{t}}{R^{2}\theta_{R}{\left(\theta_{t}R\right)}^{2}}$$
where $P_{t}$ is the laser transmitted power, $\theta_{t}$ is the laser transmitter beam diameter and angular divergence, R is the range between target and detector, dA is the effective target area, $\rho_{t}$ is the target surface reflectivity, $\theta_{R}$ is the target surface angular dispersion, $\tau_{0}$ is the transmission of the optics, $\tau_{a}$ is the atmospheric transmission, and $D_{R}$ is the area of the circular receiver aperture with diameter [18].

The parameters mentioned above are set to appropriate values, as shown in Table 1.

Figure 6a is the original array detection imaging method. For the 6 × 6 array-detection, each detector corresponds to a detection area of 100 × 100 pixels. The 36 sub-image blocks are reconstructed separately and then spliced together to obtain a 600 × 600 pixel reconstruction image.

Figure 6b is the method to obtain the reconstructed image based on the threshold. It can be seen that in the original 100 × 100 pixel block, six received target signals are higher than the threshold, and they are divided into four 50 × 50 pixel image blocks and reconstructed respectively. The remaining 100 × 100 pixel image blocks do not need to be subdivided, and all the signals received by their corresponding detectors are directly used to reconstruct the target image of the corresponding 100 × 100 pixel area. Finally, these 54 image blocks are spliced together to obtain a reconstructed image of 600 × 600 pixels.

It should be noted that the value of the length difference ΔL of the fiber bundle is the key to whether the reflected signal of the target area with low sparsity can be subdivided correctly. ΔL should ensures that the information of the four sub-imaging blocks does not interfere with each other. If ΔL is too small, the information in different areas will overlap and the reconstruction will fail. If ΔL is too large, it will waste fiber cost and increase detection time. Therefore, ΔL must satisfy:(5)ΔL>c×(ΔS/c)
where c is the speed of light, and ΔS is the depth of the target. It can be obtained that the value of ΔS should be greater than ΔS. Information for the next sub-imaging block will not reach the detector until the detector receives all the information of the previous sub-imaging block.

The quality of the reconstructed image is characterized by the difference between the reconstructed image and the original image. The higher the number of measurements, the smaller the obtained variance value, and the better the quality of the reconstructed image. Through multiple simulation and recognition, we set a variance value, which corresponds to a high-quality reconstructed image that meets the requirements of recognition. We also set the threshold of segmented pixel detection for the detector of the improved system. Figure 7a shows the intensity image of the target. Three groups of simulation experiments were organized, ten times for each group. The original and proposed system models were used to reconstruct 600 × 600 pixel image, and to calculate the average number of measurements required to obtain high-quality reconstruction results set in advance, as shown in Figure 7b.

In a parallel measurement system, the total measurement time depends on the image block with the maximum number of measurements required for reconstruction. After the reconstruction effect is set, more measurements are required, which means that the corresponding image block is more important for the overall target recognition, and may even affect the overall recognition result. Therefore, the time consumption of the two systems can be compared by comparing the peak value of the relationship curve in Figure 7b. It can be concluded that, compared with the original method, the peak value corresponding to the proposed method is significantly reduced. The extra time cost caused by the fiber array and threshold judgment only accounts for a small proportion, and becomes less as the demand for imaging pixels increases.

## 4. Threshold Selection

After verifying the advantages of the proposed system, it is considered whether the measurement time can be further shortened by adjusting the threshold. The target is placed in two positions, and three thresholds are used to reconstruct the same level of high-quality images. Figure 8a–c correspond to the first position of the target, and Figure 8d–f correspond to the second position of the target. The thresholds used in Figure 8a,d are lower than those in Figure 8b,e, and higher than those in Figure 8c,f. It can be seen from the blue and yellow blocks in Figure 8 that different thresholds result in different target area subdivision schemes. Due to the further subdivision of image blocks, the abscissa of Figure 9 corresponds to the number of sub-image blocks in Figure 8a–f, as 54, 45, 60, 51, 42, and 54, respectively.

By comparing the peaks of the curves in Figure 9a,b, it can be obtained that the number of measurements required to reconstruct a high-quality image with a larger threshold (Threshold 2) is much higher than the others. The difference between the number of measurements required for Threshold 1 and Threshold 3 is relatively small, between which the number of measurements of Threshold 1 is slightly more. However, the lower the threshold selected, the more image blocks are subdivided.

## 5. Conclusions

The single-pixel camera is able to efficiently and scalably handle high-dimensional data sets from hyperspectral imaging. Target detection and recognition have increasingly higher requirements for imaging pixels. Using a single-pixel camera to build an array-detection imaging system can greatly reduce hardware burden; however, the time cost of reconstructing images will also increase. Therefore, an improved adaptive method is proposed, which uses a flexible optical fiber array structure to connect to the detector array. The computer performs a threshold judgment on the total signals received by each detector in the detector array, and further divides the sub-image blocks that exceed the threshold (low-sparse); then, images are reconstructed separately, so the aim of shortening the reconstruction time can be achieved by parallel measurement and calculation. The comparative experiment proves the effect of the system for shortening the measurement time, and another simulation result provides suggestions for the selection of the threshold.

## Figures and Tables

**Figure 1 sensors-22-08848-f001:**
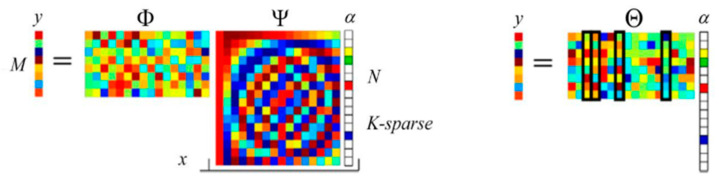
Linear measurement of CS [4].

**Figure 2 sensors-22-08848-f002:**
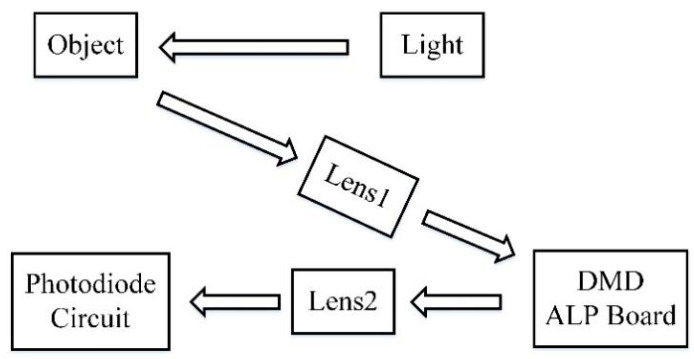
Schematic diagram of single-pixel camera.

**Figure 3 sensors-22-08848-f003:**
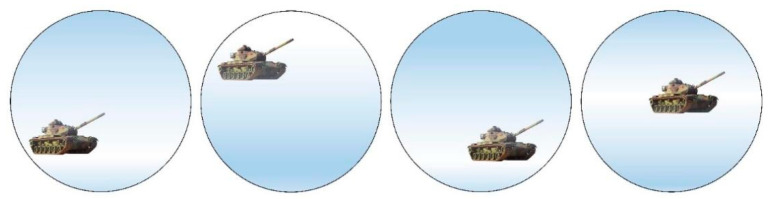
Possible target positions within the field of view.

**Figure 4 sensors-22-08848-f004:**
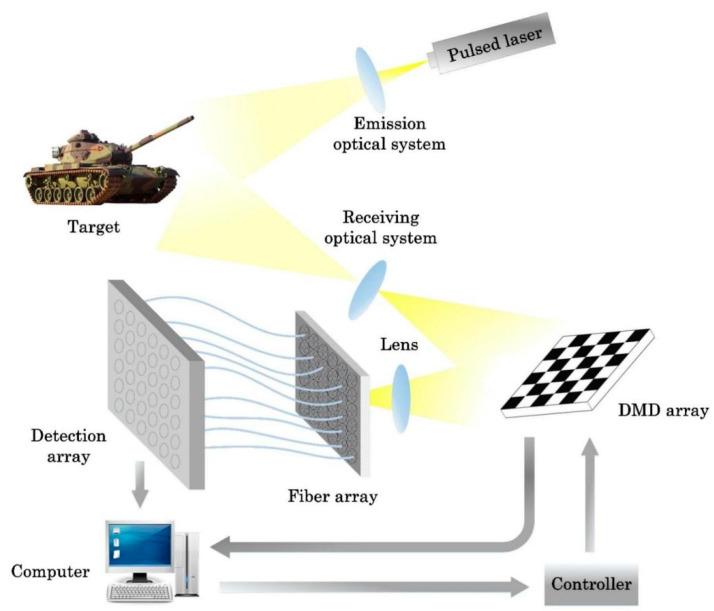
Schematic diagram of the proposed imaging system.

**Figure 5 sensors-22-08848-f005:**
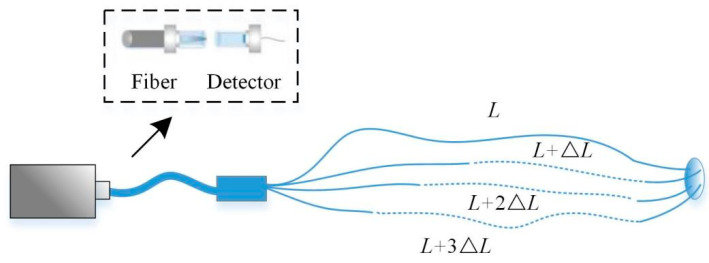
Connection structure diagram of fiber array and detector array.

**Figure 6 sensors-22-08848-f006:**
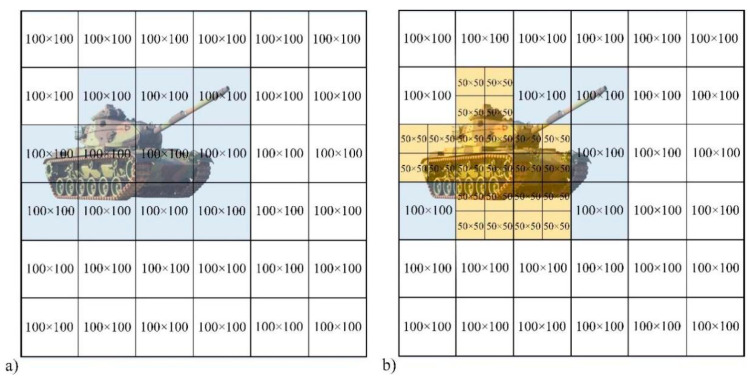
Comparison of original (**a**) and proposed (**b**) methods.

**Figure 7 sensors-22-08848-f007:**
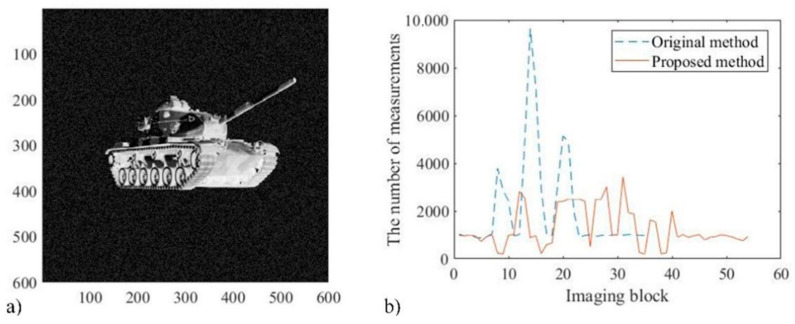
The intensity image of the target and the result of the comparison experiment.

**Figure 8 sensors-22-08848-f008:**
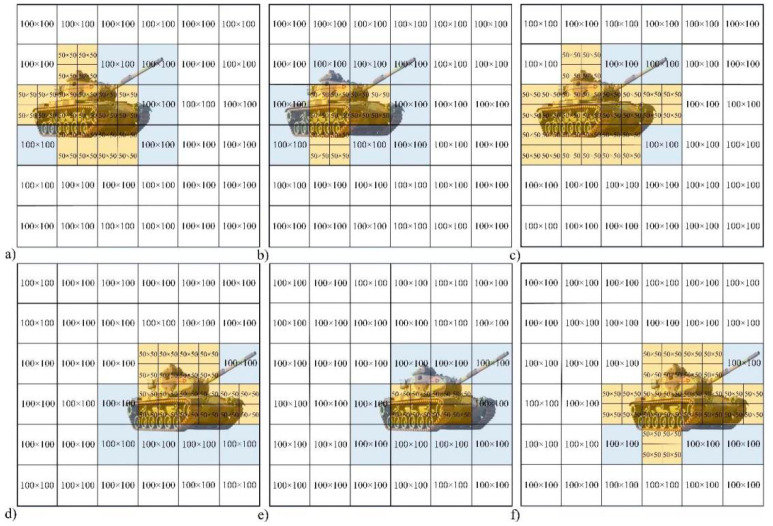
CS measurement scheme diagram with two target positions and three thresholds.

**Figure 9 sensors-22-08848-f009:**
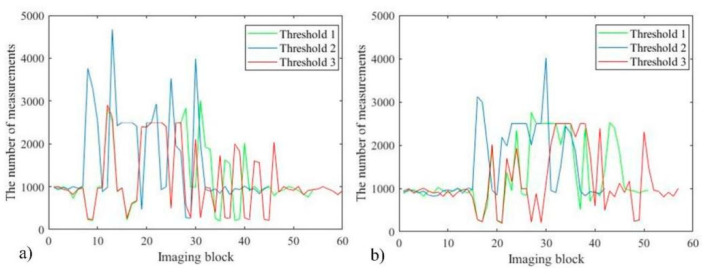
Comparison curve of the number of measurements.

**Table 1 sensors-22-08848-t001:** System simulation parameters.

Parameter	Value
laser power (J)	1
fiber diameter (μm)	125
atmospheric transmission	1
laser dispersion angular	0.012
detector dark current (A)	10^−9^
CCD pixels	800 × 800
optics transmission	1
detector quantum efficiency	0.75
width of laser pulse (ns)	6
background power (w/m^2^)	100

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
