# Peer review of "Adaptive High-Resolution Imaging Method Based on Compressive Sensing"

_sensors, 2022, doi:10.3390/s22228848_

Round 1
Reviewer 1 Report
The paper presents a method, which is adaptive regarding the image resolution and utilises the theory of compressive sensing. The proposed solution enhances existing ones and its advantages are verified experimentally. The problem is explained in details and the obtained results are discussed. I would suggest a section with literature review to be added and recently published scientific achievements to be explored. The aim of the paper should be explicitly formulated in the abstract and at the end of the Introduction section. Some typos should be corrected like: "It can be obtained that the value of S should be greater than S", etc.
Author Response
Tel: +852 61769710
Email: yfgao0502@outlook.com
Date: Oct 21, 2022
Manuscript Number: sensors-1960939
Type of manuscript: Communication
Title: Adaptive high resolution imaging method based on compressive sensing
Authors: Zijiao Wang, Yufeng Gao, Xiusheng Duan, Jingya Cao *
Dear Reviewer:
On behalf of my co-authors, thank you very much for giving us an opportunity to revise our manuscript, and we appreciate the reviewers very much for your positive and constructive comments and suggestions. We have studied reviewer’s comments carefully and have accepted all the suggestions to modify our manuscript. We would like to describe briefly the changes that we have made and present some explanations to response to the reviewer’s comments.
Thank you and best regards.
Yours sincerely,
Yufeng Gao
Answer: Thanks for your suggestion, we have updated the references and described them in the paper. The language of the dissertation's purpose has also been reinforced in the abstract and introduction, and spelling errors have been corrected.
Reviewer 2 Report
The manuscript “Adaptive high resolution imaging method based on compressive sensing” aims to take advantage of the flexibility of fiber array arrangement for fast pixel imaging. This manuscript does not make some better improvements to the algorithm or fiber, and may just add a threshold judgment on the basis of predecessors, but this is not well elaborated.
1. The author describes a lot of CS theory and single-pixel camera imaging methods throughout the paper, but they are all previous research results, and the improvement of their own research is too brief to reflect in the paper. The author should minimize the former and describe the latter in detail.
2. Regarding the detector threshold setting, the author should detail how to select the threshold, what methods are used in the selection process, and how to determine the threshold of specific data.
3. In the simulation stage, only the original system and the improved system are simulated three times, and the results are not rigorous enough because of the insufficient simulation times. The authors should increase the simulation times to ensure the repeatability and accuracy of the simulation results.
4. The author only compared the imaging results of the original system with the improved system, but a group of imaging results before and after setting the threshold should be added after the application of the fiber array. Moreover, the author only compared the measurement times of the system before and after setting the threshold value and did not specifically reflect the shortening of the measurement time mentioned by the author in the paper.
5. For target object reconstruction, the author only used one object for reconstruction in the whole process, and the sample size was too thin. At the same time, the author should add some descriptions about the image sharpness comparison after the system improvement and the reconstruction quality of moving objects.
Author Response
Dear reviewer,
Please see the attachment. Thanks.
Best regards,
Yufeng Gao

Reviewer 3 Report
The paper is not clear enough. It seems there are 36 arrays which is expensive, the background of the images is too simple, there are a lot of sentences that should be improved. A few are listed below
R. Baraniuk et al. proposed that if the relationship between the number of random measurements M and the K-sparse vectors satisfies
while sparse signals are small coefficients (such as background)
It can be obtained that the value of DS should be greater than DS
It can be concluded from the previous analysis that the more information of the target 157 is processed, the more measurement time is required for high-quality reconstruction. (Seems obvious!)
Comments on implementation are not made.
Author Response
Tel: +852 61769710
Email: yfgao0502@outlook.com
Date: Oct 21, 2022
Reply to the reviewer’s comments on our manuscript
Manuscript Number: sensors-1960939
Type of manuscript: Communication
Title: Adaptive high resolution imaging method based on compressive sensing
Authors: Zijiao Wang, Yufeng Gao, Xiusheng Duan, Jingya Cao *
Dear Reviewer:
On behalf of my co-authors, thank you very much for giving us an opportunity to revise our manuscript, and we appreciate the reviewers very much for your positive and constructive comments and suggestions. We have studied reviewer’s comments carefully and have accepted all the suggestions to modify our manuscript. We would like to describe briefly the changes that we have made and present some explanations to response to the reviewer’s comments.
Thank you and best regards.
Yours sincerely,
Yufeng Gao
Answer:
Thanks to the reviewer for the suggestion. We have reorganized the language expression to make our ideas and conclusions more clearly presented to the readers. Regarding the cost issue mentioned by the reviewer, we have started to build an experimental platform, and the fiber array has been customized, costing no more than 1,000 dollars. Compared with other devices in the system, the price is not expensive. Your suggestion is very good, we will pay more attention to the system cost while focusing on improving the recognition efficiency.
This publication focuses on the identification of targets, and only noise information is added to the background. We have begun to add occlusions and backgrounds such as trees that fit the application scene in our new research. Thank you for your suggestion.
Reviewer 4 Report
This paper designed an array-detection imaging system based on the CS technique for high-pixel detection. In the designed imaging system, each detector of the detector array is combined with a series of fibers, leading to the target area corresponding to one detector being divided into four groups of target information arriving at different times. Experimental results demonstrated the effectiveness of the designed CS-based imaging systems. However, I have the following questions:
- The reviewer cannot understand Eq. (5), the authors should give more information to explain Eq. (5).
- Why can the authors not use the proposed method to compare some existing methods?
- Some CS-based methods are missing, such as From rank estimation to rank approximation: Rank residual constraint for image restoration, Image restoration via simultaneous nonlocal self-similarity priors, Image restoration using joint patch-group-based sparse representation, Group sparsity residual constraint with non-local priors for image restoration, A benchmark for sparse coding: When group sparsity meets rank minimization.
Author Response
Dear reviewer,
Please see the attachment. Thanks.
Yours sincerely,
Yufeng Gao

Round 2
Reviewer 2 Report
1) The work is only a simulation study, which should be mentioned in the abstract.
2) Most of the cited references are relevant to CS, more relevant to imaging systems with fibers should be cited.
3) Line7, the order number of the institution should be 3.
Author Response
Dear reviewer,
Thanks for your patient and detailed suggestions, they are very useful. In the latest edition, more literature about fibers are added to the literature review, and the typo are revised. Please feel free to contact us if there is any further valuable advises.
Best regards,
yufeng
Reviewer 3 Report
I do not remember to have asked details regarding costs but perhaps background influence.
Author Response
Dear sir or Madam,
Thanks for your patient reply! The background will influence the cost.
Best regards,
Yufeng